# Fracture Toughness, Work of Fracture and Hardness of 3D-Printed Denture Base Resins

**DOI:** 10.3390/ma18184338

**Published:** 2025-09-16

**Authors:** Sebastian Hetzler, Sebastian Rehm, Sven Räther, Stefan Rues, Andreas Zenthöfer, Peter Rammelsberg, Franz Sebastian Schwindling

**Affiliations:** 1Department of Prosthodontics, Medical Faculty, Heidelberg University, Im Neuenheimer Feld 400, 69120 Heidelberg, Germany; 2Department of Prosthetic Dentistry, Medical University of Innsbruck, 6020 Innsbruck, Austria

**Keywords:** denture base, 3D-printing, fracture toughness, Vickers hardness

## Abstract

**Objectives:** To compare fracture toughness (FT), work of fracture (WOF) and Vickers hardness (HV) of four 3D-printed denture base resins—including two novel formulations—and one conventional cold-cured polymethylmethacrylate (PMMA) resin. **Methods:** 3D-printed specimens (Freeprint denture (FD)/denture impact (FDI), DETAX GmbH and V-Print dentbase/dentbase 2.0, VOCO GmbH) were fabricated at 90° layer orientation (*n* = 40/group) and notched according to ISO 20795-1. FT and WOF were measured via single-edge notched bend testing after seven-day water storage at 37 °C. HV was determined on fractured shards using 3 N load. Data were analyzed with Welch-ANOVA/Dunnett-T3 or ANOVA/Tukey (α = 0.05). **Results:** The conventional PMMA showed the highest FT and WOF, followed by the novel formulations of the 3D-printed groups VD2 and FDI. Lowest FT and WOF values were measured for VD and FD. HV was highest for the conventional PMMA, followed by the primary formulations FD and VD. Lowest hardness was measured for the novel formulations FDI and VD2. **Conclusions:** The formulations of the novel 3D-printed materials (FDI and VD2) exhibited markedly greater FT and WOF than their respective predecessors, although this improvement was accompanied by a decrease in hardness. Nevertheless, none of the 3D-printed materials fulfilled the ISO standard criteria for enhanced FT.

## 1. Introduction

Removable complete dentures are a routine treatment for edentulism, which will remain a relevant disease for the next 30 years [1]. Achieving success with removable complete dentures depends on multiple factors, including the skills of the dentist, the expertise of the dental technician, and—importantly—the materials used during fabrication. Among these, the choice of denture base material plays a critical role, with polymethyl methacrylate (PMMA) remaining the most commonly used material today.

Traditionally, PMMA-based complete dentures have been fabricated using conventional techniques, where pre-manufactured teeth are waxed up, then embedded in dental stone, de-waxed, and finally processed using hot- or cold-curing resins. Despite decades of clinical use, this technique relies on experience [2], is time-consuming, and is labor-intensive [3].

Technological advancements in digital dentistry have introduced alternative manufacturing workflows. Among these, computer-aided design and computer-aided manufacturing (CAD/CAM) have enabled the fabrication of complete dentures using subtractive milling technologies. In such workflows, denture bases are milled from pre-polymerized PMMA discs. While these materials offer superior mechanical properties [4,5], several limitations remain—including high material waste [6], and increased costs for milling equipment and materials [7].

Recent developments have seen the emergence of a third manufacturing approach: additive manufacturing, or 3D printing [8]. In contrast to the subtractive nature of milling, 3D printing builds objects layer by layer. These methods allow for a more efficient use of material, and a higher geometric freedom [9]. 3D-printing has become valuable in restorative dentistry, enabling the creation of custom dental restorations with enhanced properties. Recent advances include incorporating antimicrobial nanoparticles, such as silver and titanium dioxide, into a 3D-printed resin composite to reduce bacterial growth and improve material durability [10]. To prevent fractures and cracks like as seen in Figure 1, the mechanical properties of 3D-printed complete dentures must be critically evaluated. Recent research demonstrates that data-driven prediction methods, including graph neural networks and knowledge-based reasoning, are increasingly used to forecast and optimize mechanical performance in additive manufacturing, highlighting the importance of understanding key processing parameters and material behaviors [11,12]. This study investigated the mechanical performance of complete denture base materials fabricated using additive manufacturing techniques, with a particular focus on their fracture toughness (FT), work of fracture (WOF) and hardness (HV). The null hypotheses were that there is no difference (i) between different 3D-printed resin materials, and (ii) between 3D-printed and conventional resins regarding their FT, WOF, and HV.

## 2. Materials and Methods

### 2.1. Materials

#### 2.1.1. Sampling

Five denture base materials were investigated: four 3D-printed and one cold-cured type, with the latter serving as the reference group. All materials and the respective manufacturer are listed in Table 1.

Groups FDI and VD2 are novel variants of denture base resins, which were under development or in the process of approval at the time of testing. A sample size of *n* = 40 was used for each group, resulting in a total of 200 specimens. This number was determined by the quantity of 3D-printed samples the manufacturers could provide prior to market introduction of their materials and reflects the exploratory, pilot nature of the study.

#### 2.1.2. Sample Fabrication

The four 3D-printed groups were printed at a 90° build orientation (loaded surface with respect to the build platform) using the ASIGA MAX UV (Asiga; Sydney, Australia) with a layer thickness of 100 µm and the printing parameters recommended by the manufacturers. Post-processing was carried out according to the manufacturers’ instructions, which included ultrasonic cleaning in isopropyl alcohol and light exposure under nitrogen atmosphere with 2000 flashes from two sides (Otoflash G171, NK-Optik GmbH; Baierbrunn, Germany).

For the fabrication of the reference group AB, boards that were later sectioned into beams were cast in the first step. This was performed using the recommended mix ratio of 15 g polymer powder to 10 mL monomer. Thorough hand-mixing for 30 s and subsequent polymerization in a water bath at 40 °C and 0.2 MPa resulted in the finished boards. Sectioning of these boards was performed using a precision saw (IsoMet High Speed Pro, Buehler; Lake Bluff, IL, USA) and a diamond blade. Finally, samples were ground using #1200 grid silicone carbide paper to the final sample thickness in a semiautomatic grinding and polishing device (Tegramin25, Struers GmbH; Willich, Germany).

Corresponding to ISO 20795-1 [13], sample dimensions were 39.0 mm × 4.0 mm × 8.0 mm (±0.2 mm). All samples were measured for width and height at three spots, respectively, using a digital micrometer screw (MicroMar 40 EWR, Mahr GmbH; Göttingen, Germany) to confirm these dimensions. Samples that did not match the given size were excluded.

In a next step, a pre-crack and a sharp notch were introduced to every specimen (Figure 2a–c. Using a custom 3D-printed specimen holder, and the beforementioned precisions saw and diamond blade (0.5 mm width), a pre-crack with 3.0 ± 0.2 mm depth was inserted into three samples simultaneously (Figure 2b). Using a razor blade and a drop of glycerin, the sharp notch was then applied to each sample manually with a depth between 0.1 and 0.4 mm (Figure 2b). Figure 2c displays the representative pre-crack and sharp notch of one sample at higher magnification.

### 2.2. Fracture Toughness and Work of Fracture Testing

Prior to testing, all samples were stored in deionized water for seven days (±2 h) at 37 ± 1 °C. FT and WOF testing was performed in a three-point-bending setup according to ISO 20795-1. The notch was placed centrally under the indenter with the opening facing downwards (Figure 3a). Using a cross-head speed of 1.0 mm/min, the specimen was then loaded until fracture.

After testing, all fracture surfaces were inspected under a digital microscope (Smartzoom5, Zeiss, Jena, Germany) as seen in Figure 3b to determine the exact depths of the pre-crack and the sharp notch that were previously applied. Therefore, three separate measurements at different areas of the surface were taken and their mean value was calculated.

Using Equations (1)–(5) and Figure 4, the FT *K_max_* and WOF *W_f_* can then be determined.(1)$$K_{max}=\frac{f\cdot P_{max}\cdot l_{t}}{\left(h_{t}\cdot b_{t}^{3/2}\right)}\cdot \sqrt{10^{-3}}$$(2)$$f\left(x\right)=3x^{1/2}\cdot \frac{1.99-x\cdot (1-x)\cdot (2.15-3.93x+2.7x^{2})}{2\cdot (1+2x)\cdot {\left(1-x\right)}^{3/2}}$$(3)$$x=\frac{a}{h_{t}}$$(4)$$W_{f}=\frac{U}{\left[2b_{t}\cdot (h_{t}-a)\right]}\cdot 1000$$(5)U=∫PdΔ
with

*P_max_* = maximum applied load (N), *a* = crack-length (mm) (sum of pre-crack and sharp notch), *h_t_* = sample height (mm), *b_t_* = sample thickness (mm), *l_t_* = span between supports (32.0 ± 0.1 mm) and *U* = plotted area under the load-deflection curve (Nmm).

### 2.3. Vickers Hardness Measurement

Prior to the hardness measurements, all specimens were conditioned in deionized water at 37 ± 1 °C for 60 min. Vickers hardness measurements were conducted with shards from the fracture toughness test samples using a Duramin-40 (Struers, Willich, Germany) semiautomatic hardness tester (Figure 3c). Ten samples per group were tested, with three separate measurements on each sample and 5 mm spacings between each indentation. Each measurement was performed with a load of F = 3 N for 10 s, and the remaining indentation dimensions were measured at 40× magnification by one single operator. To calculate the Vickers hardness, the following equation was used:(6)$$HV0.3=0.1891\cdot \frac{F}{d^{2}}$$
with

F = applied load (N) and d = mean diagonal length of the indentation (mm)

### 2.4. Statistical Analysis

Statistical analysis was performed using SPSS version 29 (IBM; New York, NY, USA). Mean and standard deviation (SD) values of the recorded FT, WOF, and HV were calculated for each group and visualized by use of boxplot diagrams. Since each group of FT and WOF tests comprised 40 samples, the sampling distribution of the mean tends to approximate normality according to the central limit theorem. In the case of FT and WOF tests, group variances differed, therefore Welch-ANOVA and Dunnett-T3 pairwise post hoc tests were carried out to identify possible differences between 3D-printed and cold-cured materials. For Vickers hardness measurements, normal distribution was verified (Shapiro–Wilk). Group variances were homogeneous; therefore, ANOVA and Tukey HSD post hoc tests were carried out.

## 3. Results

### 3.1. Fracture Toughness and Work of Fracture

Table 2 and Figure 5 summarize the results of the FT and WOF tests. The cold-cured reference group AB clearly showed the highest toughness (1.44 ± 0.14 MPa∙√m) and WOF (257 ± 30 J/m^2^), followed by VD2 (0.91 ± 0.11 MPa∙√m and 140 ± 39 J/m^2^) and FDI (0.85 ± 0.12 MPa∙√m and 117 ± 35 J/m^2^) (no statistical difference between the latter two). The lowest FT and WOF values were seen for VD (0.78 ± 0.08 MPa∙√m and 69 ± 16 J/m^2^) and FD (0.69 ± 0.06 MPa∙√m and 52 ± 8 J/m^2^).

After inspection of the fracture surfaces, it was confirmed that the sharp notch was in the range between 0.1 and 0.4 mm as indicated according to ISO 20795-1.

### 3.2. Vickers Hardness

The results from the Vickers hardness measurements are summarized in Table 2 and Figure 6. The highest hardness was seen for the cold-cured reference group AB (20.9 ± 1.1 HV0.3), followed by FD (17.6 ± 0.7 HV0.3) and VD (15.8 ± 0.8 HV0.3) (no statistical difference between the latter two). The lowest hardness values were seen for FDI (12.6 ± 0.8 HV0.3) and VD2 (10.0 ± 2.2 HV0.3).

## 4. Discussion

With the exceptions of comparisons between FDI and VD2 regarding their FT and WOF and between FD and VD regarding their HV, the results differed significantly between the tested 3D-printed denture base resins. Therefore, the first null hypothesis (i) had to be partially rejected. The conventional cold-cured control material, showed significantly higher toughness and hardness compared to all 3D-printed materials. Therefore, the second null hypothesis (ii) had to be rejected. Accordingly, their risk of fracture during clinical use is lower compared to the 3D-printed materials.

The obtained FT and WOF values of the 3D-printed resins—particularly for FD and VD—are generally in line with previously reported findings of the same or comparable resins [6,14,15,16]. None of the tested materials met the requirements for improved impact resistance given in the ISO 20795-1 standard [13], which state that at least 80% of the specimen need to have a fracture toughness of ≥1.9 MPa m^½^ and/or a work of fracture of ≥900 J/m^2^. In comparison to conventional materials like cold-cured, heat polymerized or pre-polymerized CAD/CAM blocks for milling, 3D-printed denture base materials predominantly exhibit inferior FT and HV [17]. A possible explanation might be the lower degree of conversion (DC) and lower degree of crosslinking between polymer chains that is reported for 3D-printed specimen [17,18,19,20]. Another factor affecting the results could have been internal voids and pores weakening the specimen, in particular if they are located on the specimen’s periphery [6,21,22].

Regarding the DC, post-curing modalities play a significant role. A thorough post-curing can increase the DC significantly and therefore enhance mechanical properties such as flexural strength, Young’s modulus, and FT [16,20,23,24]. Choosing the correct method for each material is therefore important but may be challenging as some studies report enhanced conversion rates with post-curing by the constant emittance of UV-light for 30 min compared to using a flashlight lamp [25] (although deviant from the manufacturer’s instructions) while others report the opposite [20]. Thermal post-curing is also a viable option as increased temperatures are well known to have a considerable effect on the DC [16,26,27,28]. As described in Section 2.1, during this study, all 3D-printed samples underwent the same post-curing modality, which corresponds to the manufacturer’s instructions. This allows for a fair comparison between the materials, as would be the case in the everyday work of a commercial laboratory or dentist.

Other processing parameters can also influence the mechanical performance of 3D-printed specimen, with the most prominent examples being layer thickness, build orientation, and cleaning method [6,20,29,30]. All 3D-printed specimens in this study were fabricated at a 90° build orientation (loaded surface with respect to the build platform), which has been reported to yield the lowest mechanical properties due to the alignment of print layers along the loading axis [30,31]. This setup can therefore be seen as a worst-case scenario clinically. Results of the same denture base resin may ultimately vary between different studies, since these processing parameters can differ [14,16,30].

Significant improvements in FT and WOF demonstrated by FDI compared to FD, and by VD2 compared to VD, reflect the formulation adjustments introduced by the respective manufacturers. Although full compositional details are unavailable, insights from available safety data sheets [32,33,34] and information obtained on request from manufacturers, suggest that these property changes relate closely to variations in monomer composition—particularly the inclusion of isobornyl methacrylate (IBOMA) and the presence or absence of typical crosslinking co-monomers like Triethylenglykoldimethacrylat (TEGDMA).

The addition of IBOMA in FDI, a bulky monomethacrylate with a rigid bicyclic structure, likely reduces the overall crosslink density compared to the FD formulation [35,36,37]. This network morphology leads to decreased hardness and stiffness, as corroborated by the lower Vickers hardness observed in FDI. However, the concomitant reduction in polymerization shrinkage stress and internal residual stresses may diminish brittleness and enhance molecular mobility. These factors can promote increased energy dissipation at crack tips, thereby improving fracture toughness despite the lower hardness.

Similarly, the comparison between VD (with TEGDMA) and VD2 (without TEGDMA) supports this interpretation. The removal of TEGDMA, a low-molecular-weight, highly crosslinking dimethacrylate known to increase flexural strength and hardness [38], probably leads to a reduction in crosslink density and stiffness in VD2. While this decreases the material’s hardness, the resulting polymer network may gain ductility, contributing to an increase in FT.

A prominent approach to increase the mechanical properties of 3D-printed denture bases is to incorporate nanoparticles such as zirconia, silicon, or titania into the resin matrix [39]. Moreover, adjusting the base resin composition by adding crosslinking agents or modifying monomer ratios can be an effective way to increase the toughness and wear resistance of the final product [40].

Several limitations must be acknowledged in this study. First, the use of only one control material—cold-cured PMMA—limits the generalizability of the findings. A more comprehensive comparison including milled and heat-cured materials would offer a broader perspective. Second, only one print orientation (90°) was examined, which, although representing a worst-case scenario, does not allow conclusions on the effects of alternative nesting strategies. Third, no artificial aging or thermocycling was performed, which is essential for understanding the long-term performance of these materials. Fourth, for a profound understanding of the materials behavior, more extensive tests like the analysis of the DC, the water absorption behavior and flexural strength tests would need to be carried out. Moreover, only limited information about the chemical composition of the resins was available. To allow for an elaborate discussion about the mechanisms of improved properties, chemical analysis has to be performed in future investigations. Finally, since this in vitro investigation solely focused on the mechanical performance of the materials under a laboratory environment, only limited conclusions to the clinical performance can be drawn.

## 5. Conclusions

Within the limitations of this study, fracture toughness and work of fracture of the tested 3D-printed denture base resins were generally lower than those of the conventional cold-cured control material, which also exhibited the highest hardness. Clinicians should therefore consider these limitations when selecting materials for denture fabrication with high requirements regarding their fracture toughness. Formulation changes in the newer 3D-printed materials (FDI and VD2) resulted in significantly higher fracture toughness and work of fracture compared to their respective predecessors (FD and VD), but at the expense of reduced hardness. Further research should explore the long-term clinical performance of 3D-printed denture base resins, including the effects of aging, biocompatibility, and microbial resistance, to better predict their durability in the oral environment.

## Figures and Tables

**Figure 1 materials-18-04338-f001:**
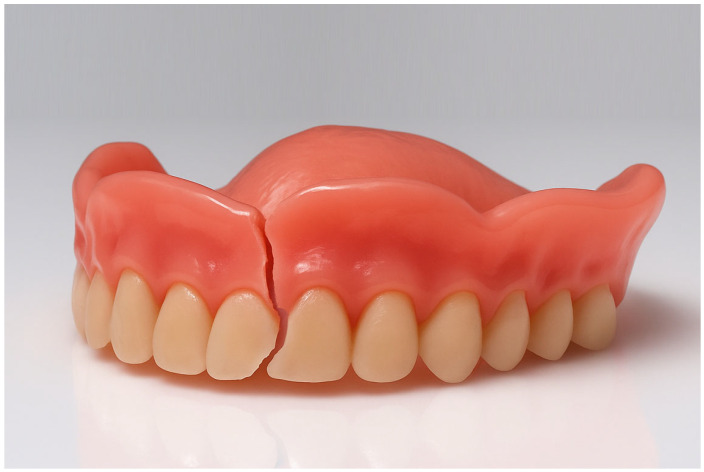
Exemplary fracture pattern commonly seen in full dentures.

**Figure 2 materials-18-04338-f002:**
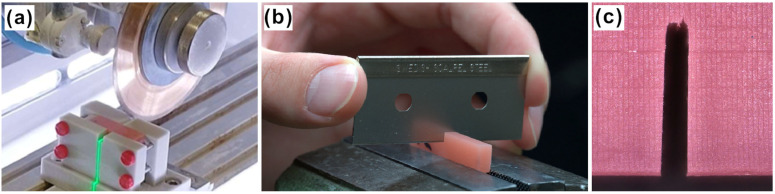
Sample production workflow. (**a**) specimen during the introduction of the notch cut using a precision saw, (**b**) specimen during the manual insertion of the blade cut, and (**c**) close-up of the notch before testing.

**Figure 3 materials-18-04338-f003:**
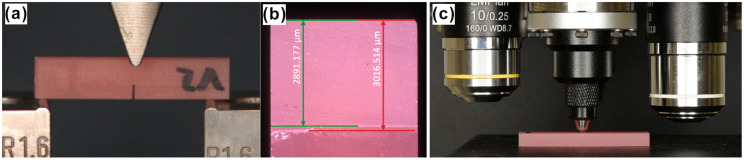
(**a**) Test setup for the fracture toughness test, (**b**) fracture surface of a specimen used for the crack length measurement, and (**c**) specimen during the Vickers hardness measurement.

**Figure 4 materials-18-04338-f004:**
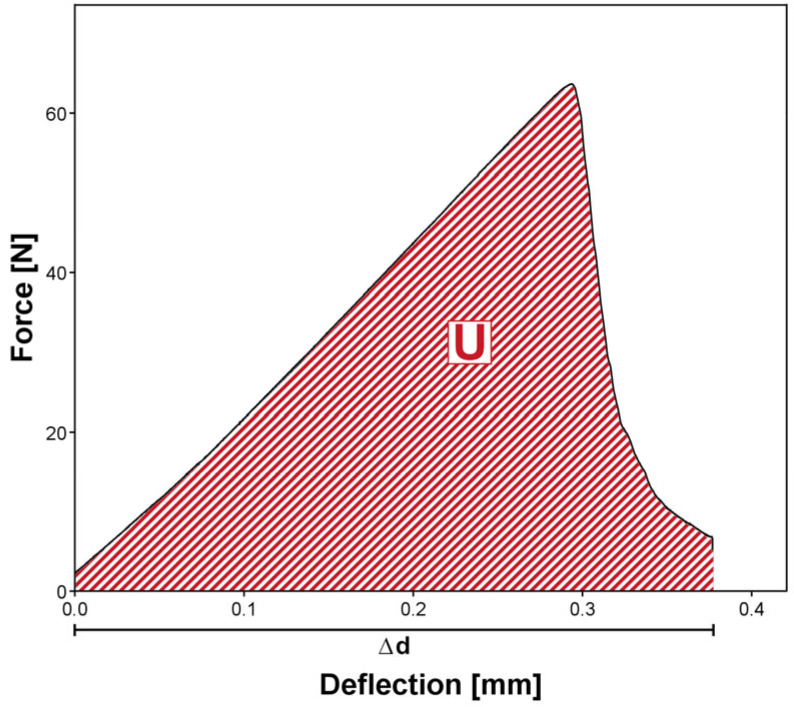
Exemplary force-deflection graph with U as the plotted area under the load-deflection curve (Nmm) and Δd as the deflection variable.

**Figure 5 materials-18-04338-f005:**
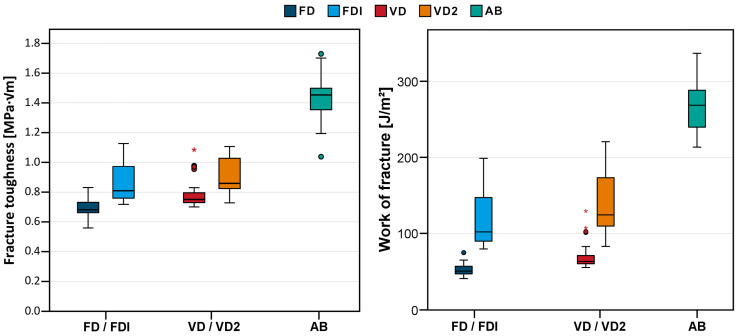
Boxplot diagram showing the fracture toughness (**left**) and work of fracture (**right**) of the test groups. FD = Freeprint denture (Detax); FDI = Freeprint denture impact (Detax); VD = V-Print dentbase (VOCO); VD2 = V-Print dentbase 2.0 (VOCO); AB = Aesthetic Blue (Candulor). Circles indicate mild outliers that lie between 1.5 and 3 times the interquartile range (IQR) outside the quartiles, while stars mark extreme outliers that are more than 3 times the IQR away from the quartiles.

**Figure 6 materials-18-04338-f006:**
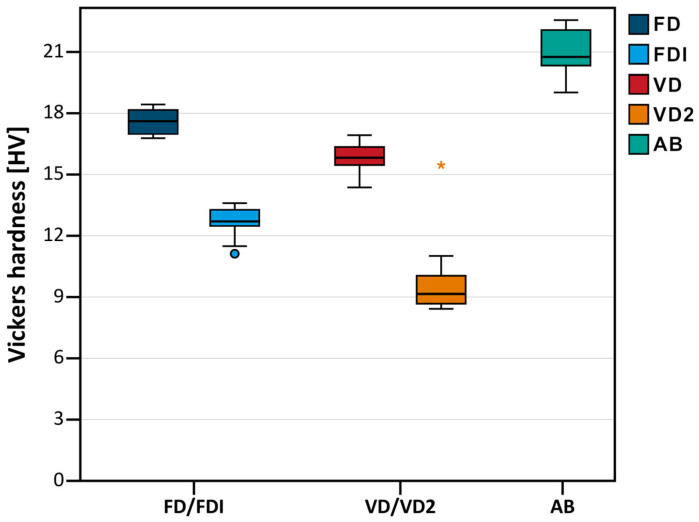
Boxplot diagram showing the Vickers hardness of the respective groups. Circles indicate mild outliers that lie between 1.5 and 3 times the interquartile range (IQR) outside the quartiles, while stars mark extreme outliers that are more than 3 times the IQR away from the quartiles.

**Table 1 materials-18-04338-t001:** Materials under investigation with additional information about the respective manufacturer and manufacturing technique.

Name	Abbreviation	Manufacturer	Technique
FREEPRINT^®^ DENTURE	**FD**	DETAX GmbH, Ettlingen, Germany	3D-Printing
FREEPRINT^®^ DENTURE IMPACT	**FDI**		
V-Print^®^ dentbase	**VD**	VOCO GmbH, Cuxhaven, Germany	
V-Print^®^ dentbase 2.0	**VD2**		
AESTHETIC BLUE^®^	**AB**	CANDULOR AG, Opfikon, Switzerland	Casting

**Table 2 materials-18-04338-t002:** Results of fracture toughness, work of fracture and Vickers hardness measurements. Different upper-case letters indicate significant differences between the test groups.

Material	Fracture Toughness [MPa∙√m]		Work of Fracture [J/m^2^]		Vickers Hardness [HV0.3]	
	Mean Value	SD	Mean Value	SD	Mean Value	SD
FD	0.69 ^A^	0.06	52 ^A^	8	17.6 ^A^	0.7
FDI	0.85 ^B^	0.12	117 ^B^	35	12.6 ^B^	0.8
VD	0.78 ^C^	0.08	69 ^C^	16	15.8 ^A^	0.8
VD2	0.91 ^B^	0.11	140 ^B^	39	10.0 ^C^	2.2
AB	1.46 ^D^	0.12	257 ^D^	30	20.9 ^D^	1.1

## Data Availability

The raw data supporting the conclusions of this article will be made available by the authors on request.

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
