# Peer review of "Fracture Toughness, Work of Fracture and Hardness of 3D-Printed Denture Base Resins"

_materials, 2025, doi:10.3390/ma18184338_

Round 1

Reviewer 1 Report

Comments and Suggestions for Authors

This study compares the fracture toughness (FT), work of fracture (WOF), and Vickers hardness (HV) of four 3D-printed denture base resins. The research is well-structured, employing standardized testing methods and robust statistical analysis. A key strength is the inclusion of next-generation 3D-printed materials (FDI and VD2) still under development, providing valuable early insights into formulation improvements. However, the study is limited by its use of only one conventional control material, a single print orientation, and the absence of aging simulations, which restricts the generalizability of the findings regarding long-term clinical performance.

Here are six key comments on the study:

  1. The study provides timely data on two new 3D-printed resins (FDI and VD2), showing significant improvements in fracture toughness and work of fracture over their predecessors. This is clinically relevant as it demonstrates manufacturers' efforts to enhance the mechanical performance of 3D-printed denture bases.

  2. The inverse relationship between fracture toughness and hardness in the novel materials is well-documented and explained. The reduction in crosslinking density (e.g., through the use of IBOMA or removal of TEGDMA) likely contributes to increased ductility and energy dissipation, albeit at the cost of surface hardness.

  3. None of the 3D-printed materials met the ISO standard requirements for improved impact resistance. This is a critical finding that underscores the current limitations of 3D-printed denture base materials compared to conventional and milled alternatives.

  4. The use of a worst-case scenario (90° build orientation) strengthens the clinical relevance of the results, as it simulates directional weakness that may occur under functional load. The sample preparation and testing procedures adhere to ISO standards, enhancing the validity of the results.

  5. The exclusion of other conventional materials (e.g., heat-cured PMMA or milled PMMA) and the lack of artificial aging limit the scope of the conclusions. Future studies should include these groups and simulate oral conditions to better predict clinical performance.

  6. The findings suggest that while newer 3D-printed resins show promise, they still lag behind conventional materials in key mechanical properties. Clinicians should consider these limitations when selecting materials for denture fabrication, especially in cases requiring high fracture resistance.

  7. please cite the following papers, if possible: [1] Graph attention-based knowledge reasoning for mechanical performance prediction of L-PBF printing parts.
    [2] Knowledge graph network-driven process reasoning for laser metal additive manufacturing based on relation mining.
    [3] Information exchange and knowledge discovery for additive manufacturing digital thread: a comprehensive literature review.
    [4] STEP/STEP-NC-compliant manufacturing information of 3D printing for FDM technology.

Overall, this study offers valuable comparative data on emerging 3D-printed denture materials and highlights both advancements and ongoing challenges in the field.

Reviewer 2 Report

Comments and Suggestions for Authors

This article deals with the current and important issue concerning the mechanical properties of new generations of 3D printing resins in prosthetics. The topic is relevant and aligns well with the current developments in digital dentistry.

The authors systematically compare classic and improved formulations, contrasting them with conventional PMMA. The manuscript is interesting and well-written. The methodology is well-described and conducted in accordance with ISO standards, and the results offer both scientific and practical value. The authors compared the properties of "classic" 3D resin formulations with their new versions (FD vs. FDI, VD vs. VD2). This demonstrates real technological progress and the potential for further development of these materials. Forty samples per group were used in the FT and WOF tests, which is a relatively large number in materials science research. This increases measurement precision and limits the influence of outlier results.

Not only fracture toughness (FT, WOF) but also hardness (HV) was tested, which provides a more comprehensive picture of the performance characteristics of the resins tested.

However, there are a few issues to be corrected/discussed:

  1. Limited control group – comparison was made with only one reference material (cold-cured PMMA), omitting other clinically important groups, such as thermopolymerized materials or materials milled from CAD/CAM blocks. This limits the generalizability of the results.
  2. Lack of material aging – no accelerated aging, thermocycling, or long-term water immersion tests were performed. This prevents the assessment of durability and stability of mechanical properties in conditions similar to those found in the oral cavity.
  3. Single printing orientation (90°) – samples were printed only in the so-called worst-case mechanical scenario (90°), which limits the possibility of drawing conclusions about other settings used in laboratory practice.
  4. Lack of additional physicochemical property testing – the authors admit that the degree of conversion (DC), water absorption, or flexural strength, among other factors, were not analyzed, which prevents a full understanding of the materials' behavior.
  5. Limited compositional information – conclusions about the mechanisms of improved properties were based primarily on data from safety data sheets and manufacturer communications, without independent chemical analysis of the resins.
  6. Lack of clinical relevance – the study focused solely on in vitro mechanical parameters, without reference to clinical performance (e.g., patient comfort, susceptibility to cracking during use).
  7. In the discussion section, please clarify the significance of the results in relation to potential clinical applications (risk of fracture during use) and compared with CAD/CAM and thermoplastic materials.
  8. Keywords, please replace „3D-pritning” for „3D-printing”

Reviewer 3 Report

Comments and Suggestions for Authors
  1. The Introduction section should consider various areas of application of 3D technologies in restorative dentistry, in particular for the creation of antimicrobial dental compositions. It is recommended to use modern research (see https://doi.org/10.3390/polym14050864, ​​doi: 10.1016/j.jdent.2024.10536).
  2. The composition of the samples (fillers, resins, etc.) should be indicated and the choice of samples for testing should be justified.
  3. References to methods for measuring fraction toughness and Vickers hardness should be provided.
  4. The requirements for the impact resistance of dental materials should be specified and methods for increasing their impact resistance should be proposed.
  5. The Conclusions section should provide quantitative data on fraction toughness and Vickers hardness for the considered samples and indicate what research is planned in the future.

Round 2

Reviewer 1 Report

Comments and Suggestions for Authors

it can be accepted